# Single-shot ultrafast imaging attaining 70 trillion frames per second

Peng Wang[1], Jinyang Liang[1,2] & Lihong V. Wang [1✉]

Real-time imaging of countless femtosecond dynamics requires extreme speeds orders of magnitude beyond the limits of electronic sensors. Existing femtosecond imaging modalities either require event repetition or provide single-shot acquisition with no more than $10^{13}$ frames per second (fps) and $3 \times 10^2$ frames. Here, we report compressed ultrafast spectral photography (CUSP), which attains several new records in single-shot multi-dimensional imaging speeds. In active mode, CUSP achieves both $7 \times 10^{13}$ fps and $10^3$ frames simultaneously by synergizing spectral encoding, pulse splitting, temporal shearing, and compressed sensing—enabling unprecedented quantitative imaging of rapid nonlinear light-matter interaction. In passive mode, CUSP provides four-dimensional (4D) spectral imaging at $0.5 \times 10^{12}$ fps, allowing the first single-shot spectrally resolved fluorescence lifetime imaging microscopy (SR-FLIM). As a real-time multi-dimensional imaging technology with the highest speeds and most frames, CUSP is envisioned to play instrumental roles in numerous pivotal scientific studies without the need for event repetition.

[1] Caltech Optical Imaging Laboratory, Andrew and Peggy Cherng Department of Medical Engineering, Department of Electrical Engineering, California Institute of Technology, 1200 East California Boulevard, Mail Code 138-78, Pasadena, CA 91125, USA. [2] Present address: Centre Énergie Matériaux Télécommunications, Institut National de la Recherche Scientifique, 1650 boulevard Lionel-Boulet, Varennes, QC J3X1S2, Canada. ✉email: LVW@caltech.edu

Cameras' imaging speeds fundamentally limit humans' capability in discerning the physical world. Over the past decades, imaging technologies based on silicon sensors, such as CCD and CMOS, were extensively improved to offer imaging speeds up to millions of frames per second (fps)[1]. However, they fall short in capturing a rich variety of extremely fast phenomena, such as ultrashort light propagation[2], radiative decay of molecules[3], soliton formation[4], shock wave propagation[5], nuclear fusion[6], photon transport in diffusive media[7], and morphologic transients in condensed matters[8]. Successful studies into these phenomena lay the foundations for modern physics, biology, chemistry, material science, and engineering. To observe these events, a frame rate well beyond a billion fps or even a trillion fps (Tfps) is required. Currently, the most widely implemented method is to trigger the desired event numerous times and meanwhile observe it through a narrow time window at different time delays, which is termed the pump-probe method[9,10]. Unfortunately, it is unable to record the event in real-time and thus only applicable to phenomena that are highly repeatable. Here, real-time imaging is defined as multi-dimensional observation at the same time as the event occurs without event repetition. It has been a long-standing challenge for researchers to invent real-time ultrafast cameras[11].

Recently, a handful of groups presented several exciting single-shot trillion-fps imaging modalities, including sequentially-timed all-optical mapping photography[12–14], frequency-dividing imaging[15], non-collinear optical parametric amplifier[16], frequency-domain streak imaging[17], and compressed ultrafast photography (CUP)[18,19]. Nevertheless, none of them has imaging speeds beyond 10 Tfps. In addition, the first three methods have their sequence depths (i.e., the number of captured frames in each acquisition) limited to <10 frames because the complexity of their systems grows proportionally to the sequence depth. One promising approach is CUP[19], which combines a streak camera with compressed sensing[18]. A standard streak camera, which has a narrow entrance slit, is a one-dimensional (1D) ultrafast imaging

device[20] that first converts photons to photoelectrons, then temporally shears the electrons by a fast sweeping voltage, and finally converts electrons back to photons before they are recorded by an internal camera (see the "Methods" section and Supplementary Fig. 1). In CUP, imaging two-dimensional (2D) transient events is enabled by a wide open entrance slit and a scheme of 2D spatial encoding combined with temporal compression[19]. Unfortunately, CUP's frame rate relies on the streak camera's capability in deflecting electrons, and its sequence depth (300 frames) is tightly constrained by the number of sensor pixels in the shearing direction.

Here, we present CUSP, as the fastest real-time imaging modality with the largest sequence depth, overcoming these barriers with the introduction of multiple advanced concepts. It breaks the limitation in speed by employing spectral dispersion in the direction orthogonal to temporal shearing, extending to spectro-temporal compression. Furthermore, CUSP sets a new milestone in sequence depth by exploiting pulse splitting. We experimentally demonstrated 70-Tfps real-time imaging of a spatiotemporally chirped pulse train and ultrashort light pulse propagation inside a solid nonlinear Kerr medium. With minimum modifications, CUSP can function as the fastest single-shot 4D spectral imager [i.e., $(x, y, t, \lambda)$ information], empowering single-shot spectrally resolved fluorescence lifetime imaging microscopy (SR-FLIM). We monitored the spectral evolution of fluorescence in real time and studied the unusual relation between fluorophore concentration and lifetime at high concentration.

## Results

**Principles of CUSP.** The CUSP system (Fig. 1a) consists of an imaging section and an illumination section. It can work in either active or passive mode, depending on whether a specially engineered illumination beam is required for imaging[12–19]. The imaging section is shared by both modes, while the illumination section is for active mode only. In the imaging section, after a

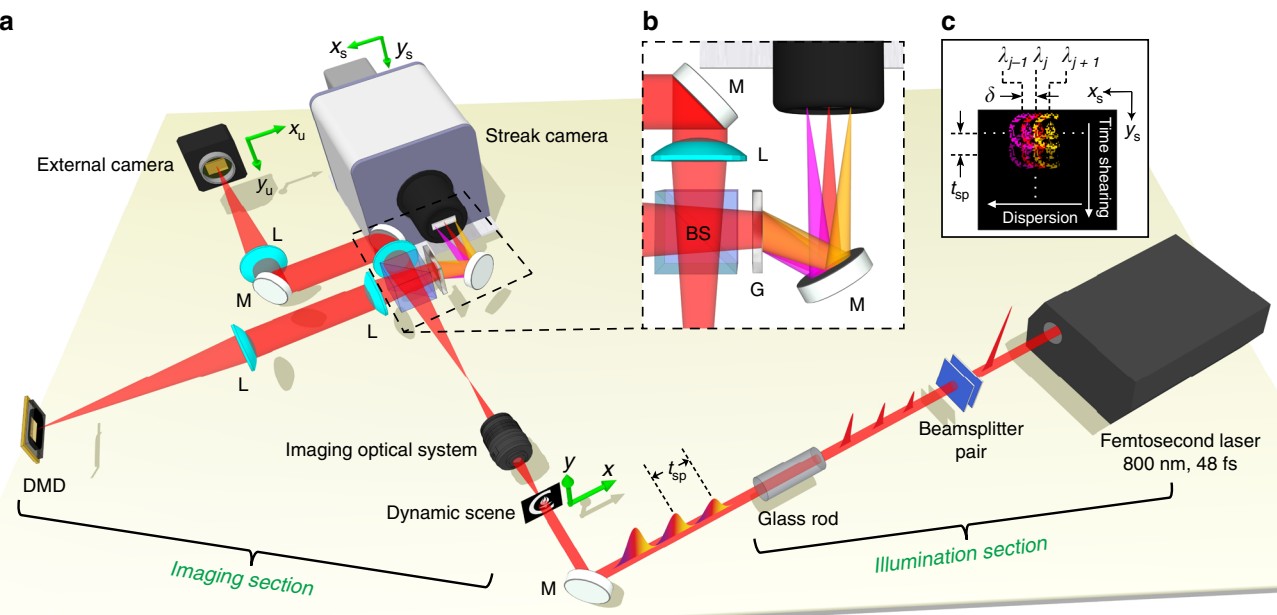

**Fig. 1 Schematic of the active CUSP system for 70-Tfps imaging. a** Complete schematic of the system. The beamsplitter pair followed by a glass rod converts a single femtosecond pulse into a temporally linearly chirped pulse train with neighboring sub-pulses separated by $t_{sp}$, which can be tuned according to the experiment. **b** Detailed illustration of the spectral dispersion scheme (black dashed box). **c** Composition of a raw CUSP image in s-View, which includes both spectral dispersion by the grating in the horizontal direction and temporal shearing by the streak camera in the vertical direction. BS, beamsplitter; DMD, digital micromirror device; G, diffraction grating; L, lens; M, mirror. Equipment details are listed in the "Methods" section.

dynamic scene $I(x, y, t, \lambda)$ is imaged by an interchangeable lens system, the light is split into two. In one path, an external camera captures a time-unsheared spectrum-undispersed image (defined as $u$-View). In the other path, the image is encoded by a digital micromirror device (DMD), displaying a static pseudo-random binary pattern, and is then relayed to the fully opened entrance port of a streak camera (see the "Methods" section). Spatial encoding by either pseudo-random[14,18,19,21] or designed[22–24] patterns is a technique extensively applied in compressed sensing. A diffraction grating, inserted in front of the streak camera, spectrally disperses the scene in the horizontal direction (see Fig. 1b). After being detected by the streak camera's photocathode, the spatially encoded and spectrally dispersed image experiences temporal shearing in the vertical direction inside the streak tube first and then spatiotemporal-spectrotemporal integration by an internal camera (see Fig. 1c). The streak camera, at the end, acquires a time-sheared spectrum-dispersed image (defined as $s$-View). See Supplementary Notes 1 and 2 for the characterizations of the streak camera and the imaging section, respectively. Retrieving $I$ from the raw images in $u$-View and $s$-View is an under-sampled inverse problem. Fortunately, the encoded recording allows us to reconstruct the scene by solving the minimization problem aided by regularization (detailed in the "Methods" section)[18,19].

In active mode, we encode time into spectrum via the illumination section, which first converts a broadband femtosecond pulse to a pulse train with neighboring sub-pulses separated by time $t_{sp}$, using a pair of high-reflectivity beamsplitters. In the following step, the pulse train is sent through a homogeneous glass rod to temporally stretch and chirp each sub-pulse. Since this chirp is linear, each wavelength in the pulse bandwidth carries a specific time fingerprint. Thereby, this pulse train is sequentially timed by $t(p, \lambda) = p t_{sp} + \eta(\lambda - \lambda_0)$, where $p = 0, 1, 2, \ldots, (P - 1)$ represents the sub-pulse sequence, $\eta$ is the overall chirp parameter, and $\lambda_0$ is the minimum wavelength in the pulse bandwidth. This timed pulse train then illuminates a dynamic scene $I(x, y, t) = I(x, y, t(p, \lambda))$, which is subsequently acquired by the imaging section. See Supplementary Note 3 for experimental details on the illumination section.

In active CUSP, the imaging frame rate is determined by $R_a = |\mu|/(|\eta| d)$, where $\mu$ is the spectral dispersion parameter of the system, and $d$ is the streak camera's pixel size. The sequence depth is $N_{ta} = P B_i |\mu|/d$, where $P$ is the number of sub-pulses, and $B_i$ is the used spectral bandwidth of the illuminating light pulse (785 nm to 823 nm). Passive CUSP does not rely on engineered illumination, and therefore, it is well suited to image various luminescent objects[11]. The independency between $t$ and $\lambda$ allows a 4D transient, $I(x, y, t, \lambda)$, to be imaged using the same algorithm without translating wavelength into time. Instead of extracting $P$ discrete sub-pulses, passive CUSP needs to resolve $N_{tp}$ consecutive frames with a frame rate of $R_p = v/d$, where $v$ is the sweeping speed of the streak tube. This calculation is based on the fact that the scene is sheared by one pixel per frame in the vertical direction ($y_s$) so that the time interval between adjacent frames is $v/d$. The number of sampled wavelengths is $N_\lambda = B_e |\mu|/d$, where $B_e$ is the bandwidth of the emission spectrum from the object. Supplementary Notes 4 and 5 contain additional information on the data acquisition model and reconstruction algorithm.

**Imaging an ultrafast linear optical phenomenon.** Simultaneous characterization of an ultrashort light pulse spatially, temporally, and spectrally is essential for studies on laser dynamics[4] and multimode fibers[25]. Here, in the first demonstration, we created a spatially and temporally chirped pulse by a grating pair (Fig. 2a). Negative and positive temporal chirps from a 270-mm-long glass

rod and the grating pair, respectively, were carefully balanced so that the combined temporal spread $t_d$ was close to the sub-pulse separation $t_{sp} = 2$ ps (detailed in Supplementary Note 6). The pulse train irradiated a sample of printed letters, which is used as the dynamic scene (see its location in Fig. 1a). Exemplary frames from CUSP reconstruction with a field of view (FOV) of 12.13 mm × 9.03 mm are summarized in Fig. 2b. See the full movie in Supplementary Movie 1. It shows that each sub-pulse swiftly sweeps across the letters from left to right. Due to spatial chirping by the grating pair, the illumination wavelength also changes from short to long over time. The normalized light intensity at a selected spatial point, plotted in Fig. 2c, contains five peaks that correspond to five sub-pulses. Each peak represents one temporal point-spread-function (PSF) of the active CUSP system. The peaks have an average full width at half maximum (FWHM) of 240 fs, corresponding to 4.5 nm in the spectrum domain (i.e., spectral resolution of active CUSP). Fourier transforming the intensity in the spectrum domain to the time domain gives a pulse with a FWHM of 207 fs, indicating that our system operates at the optimal condition bounded by the time-bandwidth limit and temporal chirp[13]. In the high-spectral-resolution regime, the Fourier-transformation relation between pulse bandwidth and duration dominates in determining temporal resolution, and thus a finer spectral resolution broadens the temporal PSF. Whereas in the low-spectral-resolution regime, temporal chirp takes over such that a poorer spectral resolution leads to a broader temporal PSF as well. Hence, there exists an optimal spectral resolution that enables the best temporal resolution[26].

Using a dispersion parameter $\mu = 23.5$ μm nm$^{-1}$, a chirp parameter $\eta = 52.6$ fs nm$^{-1}$ and a pixel size $d = 6.45$ μm, our active CUSP offers a frame rate as high as 70 Tfps. A control experiment imaged the same scene using the state-of-the-art trillion-frame-per-second CUP (T-CUP) technique with 10 Tfps[18] (see Supplementary Movie 1). Our system design allows flexible and reliable transitions between CUSP and T-CUP, as explained in Supplementary Note 7 and Supplementary Fig. 10. T-CUP's reconstructed intensity evolution at the same point exhibits a temporal spread 3.2× wider than that of CUSP. In addition, within any time window, CUSP achieves 7× increase in the number of frames compared with T-CUP (see the blue solid lines in the inset of Fig. 2c). Thus, CUSP surpasses the currently fastest single-shot imaging modality in terms of both temporal resolution and sequence depth. Figure 2d plots the reconstructed total light intensities of the five sub-pulses versus the illumination wavelength. Their profiles are close to the ground truth measured by a spectrometer.

**Imaging an ultrafast nonlinear optical phenomenon.** Nonlinear light-matter interactions are indispensable in optical communications[27] and quantum optics[28]. Optical-field-induced birefringence, as a result of third-order nonlinearity, has been widely utilized in mode-locked laser[29] and ultrafast imaging[30,31]. Here, in our second demonstration, we focused a single 48-fs laser pulse (referred to as the 'gate' pulse), centered at 800 nm and linearly polarized along the $y$ direction, into a Bi$_4$Ge$_3$O$_{12}$ (BGO) slab to induce transient birefringence, as schematically illustrated in Fig. 3a. A second beam (referred to as the 'detection' pulse)—a temporally chirped pulse train from the illumination section of the CUSP system—was incident on the slab from an orthogonal direction, going through a pair of linear polarizers that sandwich the BGO. This is a Kerr gate setup since the two polarizers have polarization axes aligned at +45° and −45°, respectively[31,32]. The Kerr gate has a finite transmittance of $T_{Kerr} = (1 - \cos \varphi)/2$ only where the gate pulse travels. Here, $\varphi$, proportional to the gate

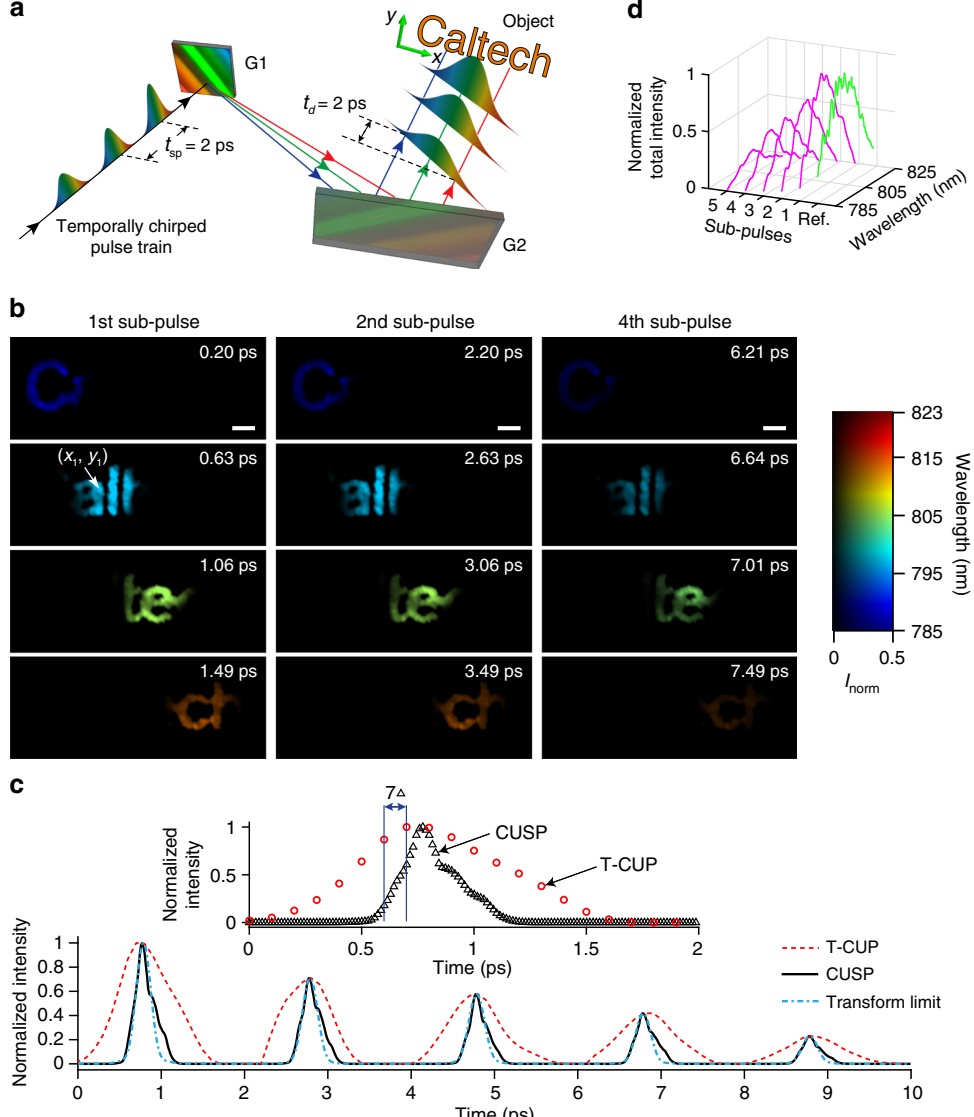

**Fig. 2 CUSP imaging of an ultrafast linear optical phenomenon. a** Schematic of the system used to impart both spatial and temporal chirps to the incoming pulse train using a pair of reflective diffraction gratings (G1 and G2). Each sub-pulse has a positive temporal chirp and illuminates the sample successively. **b** Selected frames from the reconstructed 70-Tfps movie of the spatially and temporally chirped pulse train sweeping a group of letters. Each frame is cropped from the full field of view to reduce blank space. A 2D map uses color to represent illumination wavelength and grayscale to represent intensity. Intensity is normalized to the maximum, and its color map is saturated at 0.5 to display weak intensities better. See the entire sequence in Supplementary Movie 1. Scale bars: 1 mm. **c** Temporal profile of the light intensity at $(x_1, y_1)$ (white arrow in **b**) from the state-of-the-art T-CUP (red dashed line), CUSP (black solid line) and the transform limit (cyan dash-dotted line) calculated based on the CUSP curve in the spectral domain. Inset: normalized light intensity profiles from CUSP and T-CUP in the first 2 ps. **d** Spatially integrated total intensity over wavelength for five sub-pulses (magenta lines). The reference spectrum (green line) was measured by a spectrometer.

pulse intensity, represents the gate-induced phase delay between the two orthogonal polarization directions $x$ and $y$ (see Supplementary Note 8 for its definition and measurement).

CUSP imaged the gate pulse, with a peak power density of $5.6 \times 10^{14}$ mW cm$^{-2}$ at its focus, propagating in the BGO slab. In the first and second experiments, the gate focus was outside and inside the FOV (2.48 mm × 0.76 mm in size), respectively. Figures 3b and c contain 3D visualizations of the reconstructed dynamics, which are also shown in Supplementary Movie 2. Snapshots are shown in Figs. 3d and e. As the gate pulse travels and focuses, the accumulated phase delay $\varphi$ increases, therefore $T_{Kerr}$ becomes larger. The centroid positions of the gate pulse (i.e., the transmission region in the Kerr medium) along the horizontal axis $x$ versus time $t$ are plotted at the bottom of Figs. 3b and c,

matching well with the theoretical estimation based on a refractive index of 2.07. Note that seven sub-pulses were included in the illumination to provide a 14-ps-long observation window and capture a total of 980 frames with an imaging speed of 70 Tfps. Based on the definition of $N_{ta}$, each 2-ps sub-pulse encodes 140 frames in its spectrum, and seven sub-pulses arranged in sequence offer 980 frames in total.

Re-distribution of electronic charges in BGO lattices driven by an intense light pulse, like in other solids, serves as the dominant mechanism underlying the transient birefringence[30,33], which is much faster than re-orientation of anisotropic molecules in liquids, such as carbon disulfide[14,15]. To study this ultrafast process, one spatial location from Fig. 3d is chosen to show its locally normalized transmittance evolution (Fig. 3f). Its FWHM of

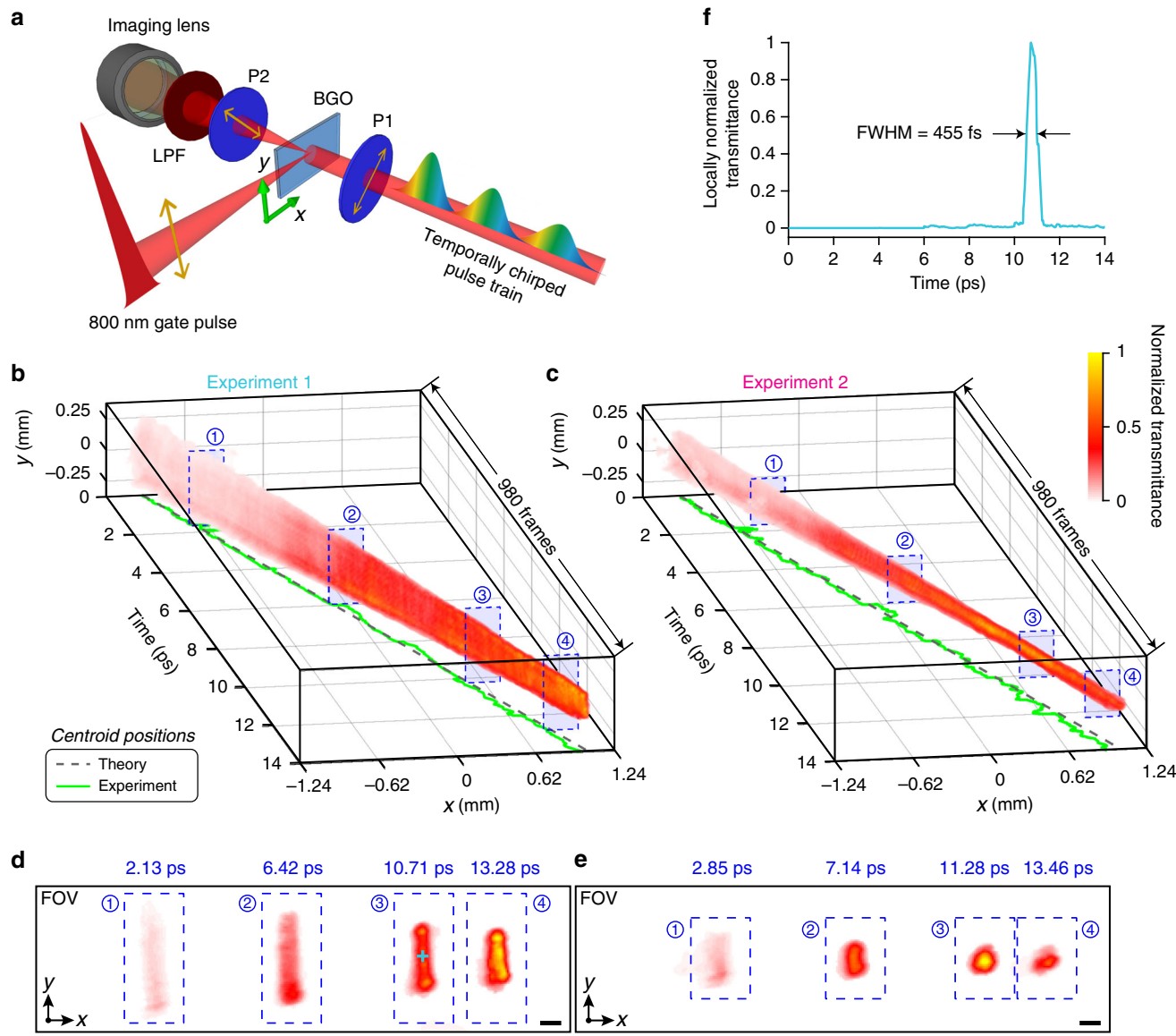

**Fig. 3 CUSP imaging of an ultrafast nonlinear optical phenomenon. a** Schematic of single-shot imaging of an ultrashort intense light pulse propagating in a Kerr medium (BGO). Polarizers P1 and P2 have polarization axes aligned at +45° and −45°, respectively. A long-pass filter, LPF, is used to block multi-photon fluorescence. A complete schematic is in Supplementary Fig. 10. **b, c** 3D visualizations of the reconstructed 980-frame movies acquired at 70 Tfps when the gate focus is (**b**) outside and (**c**) inside the field of view (FOV). Transmittance is normalized to each maximum. See the full movies in Supplementary Movie 2. The bottom surfaces plot the centroid $x$ positions of the reconstructed gate pulse versus time and the corresponding theoretical calculations. **d, e** Representative snapshots of the normalized transmittance profiles (i.e. beam profiles of the gate pulse) in the $x$–$y$ plane for **d** Experiment 1 and **e** Experiment 2, at selected time frames defined by blue dashed boxes in (**b**) and (**c**), respectively. Scale bars: 0.1 mm. **f** Evolution of the locally normalized light transmittance at a selected spatial point in Experiment 1 [cyan cross in (**d**)].

455 fs estimates a relaxation time of ~380 fs after deconvolution from the temporal PSF (Fig. 2c). This result is close to the BGO's relaxation time reported in the literature[33]. Note that T-CUP fails in quantifying this process due to its insufficient temporal resolution (see Supplementary Movie 3 and Supplementary Note 7).

In stark contrast with the well-established pump-probe method, CUSP requires only one single laser pulse to observe the entire time course of its interaction with the material in 2D space. As shown in Supplementary Note 8, the Kerr gate in our experiment was designed to be highly sensitive to random fluctuations in the gate pulse intensity, which are caused by the nonlinear relation between the Kerr gate transmittance and the gate pulse intensity. The experimental comparison in Supplementary Movie 3 reveals that the pump-probe measurement

flickers conspicuously, due to shot-to-shot variations, while CUSP exhibits a smooth transmittance evolution, owing to single-shot acquisition. Supplementary Fig. 12 shows that the fractional fluctuation in intensity is amplified 11 times in transmittance. As detailed in Supplementary Note 8, the pump-probe method would require >$10^5$ image acquisitions to capture the dynamics in Fig. 3b with the same stability as CUSP's.

**SR-FLIM.** One application of passive CUSP is SR-FLIM. Both the emission spectrum and lifetime are important properties of molecules, which have been broadly exploited by biologists and material scientists to investigate a variety of biological processes[34] and material characteristics[35]. Over the past decades, time-correlated single photon counting (TCSPC) has been the gold-

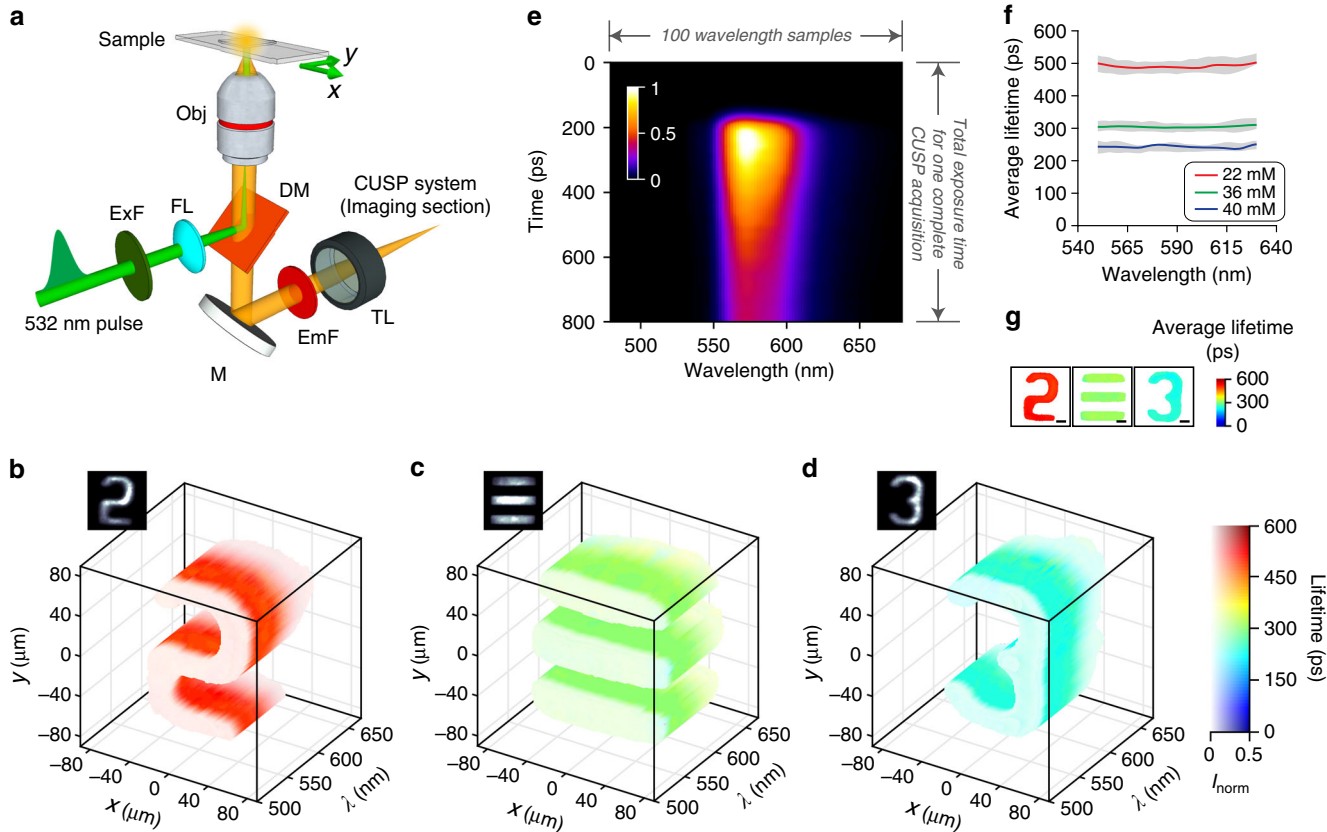

**Fig. 4 CUSP-based 4D SR-FLIM. a** Schematic of the fluorescence microscopy setup, connected with the passive CUSP system. DM, dichroic mirror; EmF, emission filter; ExF, excitation filter; FL, focusing lens; M, mirror; Obj, microscope objective; TL; tube lens. **b–d** Reconstructed lifetime distributions over the spatial (2D) and the spectral (1D) domains for three samples of different patterns and Rh6G concentrations: **b** 22 mM, **c** 36 mM, and **d** 40 mM. The entire spatial-spectral evolutions are in Supplementary Movie 4. Insets: temporally and spectrally integrated light intensity distributions in the x–y plane. **e** Spatially integrated light intensity distribution in the t–λ space for the sample of 22 mM in (**b**). This 2D distribution is normalized to its maximum. **f** Reconstructed average lifetimes at different wavelengths. The gray bands represent the standard deviations of lifetimes in the spatial domain. **g** Spectrally averaged lifetime distributions in the x–y plane for the three samples. Scale bars: 30 μm.

standard tool for SR-FLIM[36,37]. However, TCSPC typically takes tens of milliseconds to even seconds to acquire one dataset, since it depends on repeated measurements. To our knowledge, single-shot SR-FLIM has not been reported so far.

Our experimental implementation, illustrated in Fig. 4a, is a fluorescence microscope interfacing the imaging section of the CUSP system (detailed in the "Methods" section and Supplementary Note 9). This system provides a spectral resolution of 13 nm over the 200-nm bandwidth. A single 532-nm picosecond pulse was deployed to excite fluorescence from the sample of Rhodamine 6G dye (Rh6G) in methanol. Three Rh6G concentrations (22, 36, and 40 mM) with three different spatial patterns were imaged and reconstructed at 0.5 Tfps. The final data has an FOV of 180 μm × 180 μm, contains $N_{tp} = 400$ frames over an exposure time of 0.8 ns, and $N_\lambda = 100$ wavelength samples. Supplementary Movie 4 shows the reconstructed light intensity evolutions in three dimensions (2D space and 1D spectrum). Fluorescence lifetime can be readily extracted by single-exponential fitting. Figures 4b–d summarize the spatio-spectral distributions of lifetimes. Rh6G with a higher concentration has a shorter lifetime due to increased pathways for non-radiative relaxation[38]. The spatial intensity distributions (insets of Fig. 4b–d) show well-preserved spatial resolutions. Figure 4e plots the intensity distribution of the 22-mM sample in the t-λ space, clearly revealing that the emission peaks at ~570 nm and decays rapidly after excitation. Finally, we show in Fig. 4f that lifetimes remain relatively constant over the entire emission spectra and

exhibit minute variations over the spatial domain. These uniform spatial distributions are also confirmed by the spectrally averaged lifetime maps in Fig. 4g. See Supplementary Fig. 14 for more quantitative results from our SR-FLIM.

Contrary to the common observation that fluorescence lifetime is independent of concentration[3,39], our experiments demonstrate that lifetime can actually decrease with an increased concentration. Such a phenomenon was also observed in a previous study[38], attributed to the populated non-radiative relaxations. As the sample becomes highly concentrated, the fluorophores tend to stay at the excited state for a shorter time since the formations of dimers and aggregates increase pathways for relaxation (see Supplementary Note 9)[38]. To directly characterize our samples' lifetimes, we conducted a reference experiment using traditional streak camera imaging[40–42]. A uniform sample is projected onto the streak camera with a narrow entrance slit and then the lifetime is readily extracted from the temporal trace of the emission decay in the streak image. The results are plotted in Supplementary Figs. 14d and e. The difference in lifetime measurements between CUSP and the reference experiment is only 10 ps on average, much less than the lifetime.

## Discussion
CUSP's superior real-time imaging speed of 70 Tfps in active mode is three orders of magnitude greater than the physical limit of semiconductor sensors[43]. Owing to this new speed, CUSP can

quantify physical phenomena that are inaccessible using the previous record-holding system (see Supplementary Movie 3). Moreover, active CUSP captures data more than $10^5$ times faster than the pump-probe approach. When switching CUSP to passive mode for single-shot SR-FLIM, the total exposure time for one acquisition (<1 ns) is more than $10^7$ times shorter than that of TCSPC[36,37]. Additionally, CUSP is to date the only single-shot ultrafast imaging technique that can operate in either active or passive mode. As a generic hybrid imaging tool, CUSP's scope of application far exceeds the demonstrations above. The imaging speed and sequence depth can be highly scalable via parameter tuning. CUSP can cover its spectral region from X-ray to NIR[20], and even matter waves such as electron beams[24], given the availability of sources and sensing devices. In addition, CUSP is advantageous in photon throughput, compared with existing ultrafast imaging technologies[11].

Both the pump-probe and TCSPC methods require event repetition. Consequently, these techniques are not only slower than CUSP by orders of magnitude as aforementioned, but are also inapplicable in imaging the following classes of phenomena: (1) high-energy radiations that cannot be readily pumped such as annihilation radiation (basis for PET)[44], Cherenkov radiation[45], and nuclear reaction radiation[6]; (2) self-luminescent phenomena that occur randomly in nature, such as sonoluminescence in snapping shrimps[46]; (3) astronomical events that are light-years away[44,47]; and (4) chaotic dynamics that cannot be repeated[48,49]. Yet, CUSP can observe all of these phenomena. For randomly occurring phenomena, the beginning of the signal can be used to trigger CUSP. Large amounts of theoretical and technical efforts are required before CUSP can be widely adapted for these applications.

A trade-off between imaging speed and recording time is always found in single-shot ultrafast imaging modalities[7,8,12–19,24–26]. Due to the vast discrepancies in imaging speeds, it is inconvenient to compare different approaches using recording time. Therefore, we introduced a parameter for fair comparison – sequence depth[11], which is independent of imaging speeds. In CUSP with a fixed hardware configuration, the maximum sequence depth is solely determined by the FOV [see Eqs. (5) and (6) in the "Methods" section]. In a practical setting, active CUSP can acquire more than $10^3$ frames in one snapshot, which is several times[7,18,19] or even orders of magnitude[8,12–17] higher than those achievable by the state-of-the-art techniques (see the "Methods" section). Additionally, this interplay between sequence depth and FOV limits the maximum FOVs for scenarios where the desired sequence depths are given [see Equations (7) and (8) in the "Methods" section].

Compressed-sensing-enabled single-shot imaging typically has to pay the penalty of compromised spatial resolutions[14,18,21,22,50] that are caused by the multiplexing of multi-dimensional information and finite encoding pixel size. A characterization experiment (see Supplementary Note 10 and Supplementary Fig. 15) suggests a 2–3× degradation in spatial resolutions in CUSP. Advanced concepts, such as lossless encoding[7] and multi-view projection[51], are promising in compensating for this resolution loss. The success of compressed sensing relies on the premise that the unknown object is sparse in some space. For CUSP, this precondition is justified via calculations in Supplementary Table 1. All the scenes in this work have data sparsity >90%, which is sufficient to lead to satisfactory reconstructions, according to a former study[52]. This restriction on sparsity may be alleviated by optimizing the encoding mask[23,53] or the regularizer[54] in Equation (2).

CUSP's temporal resolution in active mode is essentially time-bandwidth limited[13]. Alternative encoding mechanisms, such as polarization encoding, may be explored to break this barrier and boost the imaging speed to the $10^{15}$-fps regime. The imaging speed of passive CUSP is confined by the streak camera. In the past, streak camera technology has been a powerful workhorse in science and engineering[20,40–42,55]. However, it suffers from the low quantum efficiency, space-charge effect, and intrinsic electronic jitter. Most importantly, boost in its imaging speed is at the mercy of the development of faster electronics. We envision that optical shearing that avoids the complex photon–electron conversions could bring forth the next leap in ultrafast imaging. Recent progress in machine learning may facilitate image reconstruction[56,57]. All of these directions represent only a small portion of the overall efforts in pushing the boundaries of ultrafast imaging technologies.

## Methods

**Setups and samples.** In the imaging section of the active CUSP system (Fig. 1a), the light path of s-View starts by routing the intermediate image, formed by the interchangeable imaging system, to the DMD by a 4f imaging system. A static pseudo-random binary pattern with a non-zero filling ratio of 35% is displayed on the DMD. The encoded dynamic scene is then relayed to the entrance port of the streak camera via the same 4f system. In order to maximize photon utilization efficiency of s-View, a dual-projection scheme was adopted instead in the passive CUSP system for SR-FLIM (see Supplementary Note 9 for details). In the experiment in Fig. 2a, the group of letters were printed on a piece of transparency film to impart complex spatial features. The BGO slab (MTI, BGO12b101005S2) used in the experiments in Fig. 3 has a thickness of 0.5 mm. Its edges were delicately polished by a series of polishing films (Thorlabs) to ensure minimal light scattering for optimal coupling of the gate pulse. In the fluorescence microscopy setup in Fig. 4a, a single excitation pulse was focused on the back focal plane of an objective to provide wide-field illumination in the FOV. The Rh6G solution was masked by a negative USAF target, placed at the sample plane. A dichroic mirror and a long-pass filter effectively blocked stray excitation light. After a tube lens, the image was directed into the imaging section of the passive CUSP system.

**Equipment.** The imaging section in Fig. 1a includes a 50/50 non-polarizing beamsplitter (Thorlabs, BS014), a DMD (Texas Instruments, LightCrafter 3000), an external CCD camera (Point Grey, GS3-U3-28S4M), 150-mm-focal-length lenses (Thorlabs, AC254-150-B), a 300 lp mm$^{-1}$ transmissive diffraction grating (Thorlabs, GTI25-03), and a streak camera (Hamamatsu, C6138). In the illumination section, a 270-mm-long N-SF11 glass rod (Newlight Photonics, two SF11L1100-AR800, SF11G1500-AR800, combined with SF11G1200-AR800) was used for the experiment in Fig. 2, while a 95-mm-long N-SF11 glass rod (Newlight Photonics, SF11G1500-AR800, SF11G1400-AR800, combined with SF11G1050-AR800) was used for the experiments in Fig. 3. A femtosecond laser (Coherent, Libra HE) was used as the light source in active CUSP. A picosecond laser (Huaray, Olive-1064-1BW) was used in passive CUSP for fluorescence excitation.

In Fig. 2a, the pair of 300 lp mm$^{-1}$ reflective gratings are G1 (Newport, 33025FL01-270R) and G2 (Thorlabs, GR25-0310). Figure 3a includes two linear polarizers P1 (Thorlabs, LPVIS100-MP2), P2 (Newport, 05P109AR.16), and a long pass filter (Thorlabs, FGL715). The imaging optics for the experiment in Fig. 2 includes two lenses (Thorlabs, AC508-100-B, and AC127-025-B), giving a ×0.25 magnification. A pair of lenses (Thorlabs, AC254-125-B, and AC254-100-B), with a magnification of ×0.8, was used for the experiments in Fig. 3. The SR-FLIM setup, shown in Fig. 4a, consists of a dichroic mirror (Thorlabs, DMLP550R), a long-pass emission filter (Thorlabs, FEL0550), a short-pass excitation filter (Thorlabs, FES0600), a 75-mm-focal-length focusing lens (Thorlabs, LA1608), a ×4 infinity-corrected objective (Olympus, RMS4X), and a 200-mm-focal-length tube lens (Thorlabs, ITL200).

The DMD spatially encodes the scene by turning each micromirror to either +12° (ON) or –12° (OFF) from the DMD's surface normal. Each micromirror has a flat metallic coating and reflects the incident light to one of the two directions. Therefore, we can collect the encoded scene either in a retro-reflection mode by tilting the DMD by 12°, which is used in active CUSP (Fig. 1a), or by using two separate sets of relay optics, which is used in passive CUSP (see Supplementary Fig. 13a). Here, one DMD code has a lateral dimension of 56.7 μm × 56.7 μm. Thus, the relay optics with ~0.08 NA has enough spatial resolution to image the DMD onto the streak camera.

As schematically detailed in Supplementary Fig. 1, in the streak camera, the image of the fully opened entrance slit is first relayed to the 3-mm-wide photocathode by the input optics. Then the photocathode converts photons into photoelectrons, which are accelerated through an accelerating mesh. The streak camera can operate in two modes: focus mode and streak mode. In the focus mode, no sweep voltage is applied so that an internal CCD camera (Hamamatsu, Orca R2) only captures time-unsheared images, akin to a conventional sensor. In the streak mode, these photoelectrons experience a temporal shearing on the vertical axis driven by an ultrafast linear sweep voltage. The highest sweeping speed is 100 fs per pixel, equivalently 10 THz[18,20]. The photoelectron current is amplified by a microchannel plate via the generation of secondary electrons. After a phosphor

screen converts the electrons back to photons, an internal CCD camera captures a single 2D image of the photons.

**Image acquisition and reconstruction.** We denote the optical energy distributions recorded in $u$-View and $s$-View as $E_u$ and $E_s$, which are related to the dynamic scene $I(x, y, t, \lambda)$ by

$$\begin{bmatrix} E_u(x_u, y_u) \\ E_s(x_s, y_s) \end{bmatrix} = \begin{bmatrix} T Q_u F_u \\ \alpha T S_t Q_s S_\lambda D F_s C \end{bmatrix} I(x, y, t, \lambda), \qquad (1)$$

where $C$ represents the spatial encoding by the DMD; $F_u$ and $F_s$ describe the spatial low-pass filtering due to the optics in $u$-View and $s$-View, respectively; $D$ represents image distortion in $s$-View with respect to the $u$-View; $S_\lambda$ denotes the spectral dispersion in the horizontal direction; $Q_u$ and $Q_s$ are the quantum efficiencies of the external CCD and the photocathode of the streak camera, respectively; $S_t$ denotes the temporal shearing in the vertical direction; $T$ represents spatiotemporal-spectrotemporal integration over the exposure time of each CCD; and $\alpha$ is the experimentally calibrated energy ratio between the streak camera and the external CCD. Here, we generalize the intensity distributions of the dynamic scenes observed by both active CUSP and passive CUSP as $I(x, y, t, \lambda)$ for simplicity. The concatenated form of Eq. (1) is $E = OI(x, y, t, \lambda)$, where $E = [E_u, \alpha E_s]$ and $O$ stands for the joint operator.

By assuming the spatiotemporal-spectrotemporal sparsity of the scene and calibrating for $O$, $I(x, y, t, \lambda)$ can be retrieved by solving the inverse problem defined as[18,58]

$$\arg\min_I \left\{ 0.5 \| E - OI \|_2^2 + \xi \Phi(I) \right\}. \qquad (2)$$

In Equation (2), argmin represents the argument that minimizes the function in the following bracket. The first term denotes the discrepancy between the solution $I$ and the measurement $E$ via the operator $O$ and $\| \cdot \|_2$ represents $L^2$ norm. The second term enforces sparsity in the domain defined by the following regularizer $\phi(I)$ while the regularization parameter $\xi$ balances these two terms. We opt to use total variation (TV) in the four-dimensional $x$–$y$–$t$–$\lambda$ space as our regularizer. For an accurate and stable reconstruction, a software program adapted from the two-step iterative shrinkage/thresholding (TwIST) algorithm[58] was utilized. More details on data acquisition and reconstruction can be found in Supplementary Notes 4 and 5. In addition, the assumption of data sparsity is justified in Supplementary Table 1.

CUSP's reconstruction of a data matrix of dimensions $N_x \times N_y \times N_{ta}$ (active mode) or $N_x \times N_y \times N_{tp} \times N_\lambda$ (passive mode) requires a 2D image of $N_x \times N_y$ in $u$-View and a 2D image of $N_{col} \times N_{row}$ in $s$-View. In active mode,

$$N_{col} = N_x + (N_{ta}/P) - 1, \qquad (3.1)$$

$$N_{row} = N_y + \left( v t_{sp}/d \right) \times P - 1; \qquad (3.2)$$

in passive mode,

$$N_{col} = N_x + N_\lambda - 1, \qquad (4.1)$$

$$N_{row} = N_y + N_{tp} - 1. \qquad (4.2)$$

In Equations (3) and (4), $P$ is the number of sub-pulses, $v$ is the streak camera's shearing speed, $t_{sp}$ is the temporal separation between adjacent sub-pulses, and $d$ is the streak camera's pixel size. The finite pixel counts of the streak camera ($N_x^{sc} \times N_y^{sc} = 672 \times 512$ after $2 \times 2$ binning) physically restrict $N_{col} \leq 672$ and $N_{row} \leq 512$. In active CUSP imaging shown in Fig. 2, a raw streak camera image of $609 \times 449$ pixels was required to reconstruct a data matrix of $N_x \times N_y \times N_{ta} = 470 \times 350 \times 700$. Similarly, in Fig. 3, reconstruction of a data matrix of $N_x \times N_y \times N_{ta} = 310 \times 90 \times 980$ requires an image of $449 \times 229$ pixels from the streak camera. Note that here $N_{ta}/P$ equals the number of wavelength samples within one sub-pulse, which is 140 in our active CUSP system. In the passive-mode imaging shown in Fig. 4, an SR-FLIM data matrix of $N_x \times N_y \times N_{tp} \times N_\lambda = 110 \times 110 \times 400 \times 100$ is associated with a streak camera image of $209 \times 509$ pixels.

**Limits in sequence depth and FOV.** Based on Equations (3) and (4) above, when assuming fixed FOVs, there are upper limits in the sequence depth achievable by CUSP. It is straightforward to derive this limit for active mode:

$$N_{ta}^{max} = 140 \cdot round\left[ \left( N_y^{sc} - N_y + 1 \right) \cdot d / \left( v t_{sp} \right) \right]. \qquad (5)$$

In Eq. (5), 140 represents the number of wavelength samples in one sub-pulse. $N_{ta}^{max}$ is fundamentally determined by how many sub-pulses ($P$) can be accommodated in temporal shearing. For passive mode, the maximum sequence depth is simply

$$N_{tp}^{max} = N_y^{sc} - N_y + 1. \qquad (6)$$

Taking practical numbers for example, when using the 70 Tfps active CUSP to image a scene of $N_x \times N_y = 350 \times 350$ pixels, we can obtain a maximum sequence depth $N_{ta}^{max} = 1120$ frames. This number is more than 3 times the maximum sequence depth obtained in T-CUP[18] and more than 18 times that of the best

single-shot femtosecond imaging modality other than CUP[17]. The recording time in this case is 16 ps. For the 0.5 Tfps passive CUSP, if the scene is $N_x \times N_y = 100 \times 100$, then $N_{tp}^{max} = 413$ frames, which corresponds to a recording time of 826 ps.

Similarly, for scenarios with both sequence depth and number of wavelength samples fixed, the streak camera limits the spatial pixel counts in the reconstructed data in active mode to be $N_x \leq N_x^{sc} - (N_{ta}/P) + 1$ and $N_y \leq N_y^{sc} - \left( v t_{sp}/d \right) \times P + 1$. In passive mode, these limits are $N_x \leq N_x^{sc} - N_\lambda + 1$ and $N_y \leq N_y^{sc} - N_{tp} + 1$. Considering the camera pixel size $d$ and the magnification of the imaging optics $M$, we can obtain the maximum FOV for active mode:

$$FOV_x^{max} = \left[ N_x^{sc} - (N_{ta}/P) + 1 \right] \cdot (d/M), \qquad (7.1)$$

$$FOV_y^{max} = \left[ N_y^{sc} - \left( v t_{sp}/d \right) \times P + 1 \right] \cdot (d/M), \qquad (7.2)$$

and for passive mode:

$$FOV_x^{max} = \left( N_x^{sc} - N_\lambda + 1 \right) \cdot (d/M), \qquad (8.1)$$

$$FOV_y^{max} = \left( N_y^{sc} - N_{tp} + 1 \right) \cdot (d/M). \qquad (8.2)$$

Inserting practical numbers into Equations (7) and (8) and using the configurations of our three demonstrations (Figs. 2–4), we have the maximum FOVs of 13.75 mm × 10.66 mm, 4.30 mm × 3.00 mm, and 0.92 mm × 0.18 mm, respectively.

## Data availability

The data that support the findings of this study are available from the corresponding author on reasonable request.

## Code availability

The reconstruction algorithm is described in detail in Methods and Supplementary Information. We have opted not to make the computer code publicly available because the code is proprietary and used for other projects.

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

## Acknowledgements

The authors thank Dr. Liren Zhu for assistance with the reconstruction algorithm and Dr. Junhui Shi for providing the control program of the precision linear stage. This work was supported in part by National Institutes of Health grant R01 CA186567 (NIH Director's Transformative Research Award).

## Author contributions

P.W. conceived the system design, built the system, performed the experiments, developed the reconstruction algorithm and analyzed the data. J.L. contributed to the early stage development and experiment. L.V.W. initiated the concept and supervised the project. All authors wrote and revised the paper.

## Competing interests

The authors disclose the following patent applications: WO2016085571 A3 (L.V.W. and J.L.), U.S. Provisional 62/298,552 (L.V.W. and J.L.), and U.S. Provisional 62/904,442 (L.V.W. and P.W.).
