## [Peer Review File · Nature Communications]

Reviewers' comments:

Reviewer #1 (Remarks to the Author):

It is very interesting paper and the technique is very novel. The developed compressed ultrafast spectral photography (CUSP) provides in active mode 7×10^{13} fps and passive mode 4D spectral imaging at 0.5×10^{12} fps. The spectrally resolved FLIM mode is compared with time-resolved single photon counting (TCSPC).

There are few points may be explained in this paper;

1. One of the important parameters is the recording time. What is the maximum recording time for this technology? Usually there is an inherent trade-off between longer sweep windows (recording time) and the time resolution. Authors have to explain this issue in their discussion section. If the recording time is very short, it is impossible to obtain 70 trillion frames in one second in the real world.
2. In the discussion, it is better to add few sentences about the limitations of the technology. This current technology is probably only limited to relatively stationary samples (compared to the trillion frames per second speed). If your sample is moving (or shape changing) at the speed of your frame rate, is it possible to capture the above mentioned changes? Although the majority of samples do not move that fast (they are stationary sample in this time scale).
3. The authors should have used solvent effect to measure the lifetime testing rather than the concentration variation in Rh6G. The concentration used was very high so low lifetime values were recorded. Any new development should be compared with the existing literature value <https://pubs.acs.org/doi/pdf/10.1021/j100535a016>

In addition, they have not compared the measured lifetime with the TCSPC or any other existing methods to prove that their measurement is reproducible with the literature value.

Reviewer #2 (Remarks to the Author):

The manuscript by Wang et al reports on a photography technique capable of capturing a large number of frames at a high frame rate of a single event in the femtosecond regime. The authors break records in number of frame rate and number of frames. There is no doubt that the experiment is nice and well done.

But:

- the manuscript is in general extremely emphatic (which is quite annoying) without explaining the science behind. As an example: the abstract even doesn't mention the key concept which enabled the record. Other example : "On promising approach is CUP which creatively [!] combines streak camera and compressed sensing" (with 2 references to the authors own work) . Identically, the conclusion extends the possibilities of this techniques to fields that are quite exaggerated: even I must admit that it might be possible one day, the path is quite long to detect gravitational waves from colliding black holes using CUSP...

-as a consequence of this lack of simple scientific explanations, it is to me hard to find where is the actual new concept in comparison with reference 14 (Lu et al, Phys. Rev Lett 2019 Figure 1c.)

- The authors always avoid quantitative comparisons. As a scientist, I know that nothing is magic and that incredible numbers here are always linked to some losses there (in resolution, information,...etc). It might be acceptable, but the manuscript doesn't describe them at all. Apart from the records numbers, it is in general quite difficult to get factual numbers or to see from where they are coming from. Examples: what are the temporal and spatial resolutions ? The authors seem to always avoid this point:

i /it is never provided in the main text

ii/ in Supplementary material page 3 "with a moderate SNR, the temporal resolution is typically larger than 400 fs [2]. Fortunately, the temporal resolution of active CUSP is no longer bounded by this limit." : so... what is it?

iii/ In Suppl. Mat page 16 "After CUSP reconstruction, we spatially integrated the reconstructed intensity and plotted it in the t - λ space (Supplementary Fig. S13a). It is the impulse response of the system, or in other words, the 2D PSF. The 1D PSFs in the spectral and time domains give FWHMs of 13 nm and 20 ps, respectively". So that the reader has the 1D PSF in the case of passive CUSP. It is not mentioned for the interesting (central in the manuscript) case of the active one. (Besides, I hope that it is much shorter than 20 ps....)

-What is the field of view? How is it limited?

- last but not least example is to obtain 980 frames with only 7 pulses. This number of 980 would largely deserve some lines of comments to explain how it has been obtained and if it actually means something physically. (pixelation vs resolution)

- In average, the main text is not understandable by itself without reading the Supplementary material. As an example, the actual role of the DMD is not described in the main text and barely mentioned in the suppl. mat.

Other Comments following the text:

-line 67: it is hard to see what are these passive/active modes before understanding the initial concept.

-line 103: I do not see how the frame rate R_p is calculated since each 2D frame is not with a single pixel.

-lines 119-122 are not understandable without reading ref 13

- how is implemented T-CUP in the experiments? (removal of the grating?)

-line 129: I do not see how the 7x increase comes from the figure.

-fig3a: why is P1 at 45° from the polarization axis of the Kerr gate? (it should be in the same direction?)

-line 157: how are obtained the 980 frames? I do not see how this number can be obtained with only 7 laser pulses. Are they "physically" independent? ie separated by the temporal PSF ?

-line 158: the long paragraph is nice, but the only number which would interesting (the relaxation time that could be compared with the experimental data) is not provided.

-line 172: why an amplification of the noise by 11 times for a Kerr process? Comment is needed.

-line 191: the lifetime should be provided quantitatively and compared to literature on a quantitative basis.

Methods:

-Can the sparsity be justified ?

-What is the exact mathematical mean of the argmin function and the kind of norm 2 ? (these should be specified clearly)

Supplementary materials:

-In general, the supplementary document strongly lacks of referencing. Yet it is detailed, it would be good to know how the formulas have been established (example: equation S2 for the transformation D).

- it would be help to tell the reader where the characterization have been performed without the DMD and retrieval algorithm (if so)

- Spectral sensitivity : the authors assume uniform transmission of the optics etc, but in fact, this should be already taken into account in the quantum efficiency of the apparatus.

- What is the NA of the optics? What is the resolution necessary to image the DMD?

- What is "rect" function precisely standing for?

- Page 9: it would be good to make the link with the operator O used in the main text

- What is precisely t_{sp} ?

- Page 11: how the regularization parameters were chosen? Impact on results?

- Page 17: order of magnitude for the decay? (so that one can compare to the results)

Reviewer #3 (Remarks to the Author):

This paper proposes a new ultrafast imaging technique that significantly extends the previous state of the art in several directions. The proposed method leverages several techniques (namely compressive sensing, spectral encoding, pulse splitting, and temporal shearing) to achieve impressive performance in ultrafast imaging. Moreover, the proposed approach is flexible in that it allows a tradeoff between speed and spectral resolution, making it also an ultrafast multispectral imaging technique. There are many applications for such fast imaging capabilities, thus these advances are not only technically, but also scientifically very relevant. As far as I am aware, this is a qualitative jump in the field of ultrafast imaging, thus it is an important contribution to the toolbox of science.

The paper is well written, the technique clearly motivated and explained, thus I believe that this paper is worthy of publication in its current form.

Response to Reviewers

We thank the reviewers for their insightful comments, which have helped us improve the quality of our manuscript. Below, we provide point-by-point responses to all the comments (shown in blue). The changes in the text are highlighted in red.

Reviewer 1

Comment 1.1. It is very interesting paper and the technique is very novel. The developed compressed ultrafast spectral photography (CUSP) provides in active mode 7×10^{13} fps and passive mode 4D spectral imaging at 0.5×10^{12} fps. The spectrally resolved FLIM mode is compared with time-resolved single photon counting (TCSPC).

[Response]: We thank the reviewer for supporting our work and acknowledging that our work is *very interesting* and is *very novel*.

Comment 1.2. One of the important parameters is the recording time. What is the maximum recording time for this technology? Usually there is an inherent trade-off between longer sweep windows (recording time) and the time resolution. Authors have to explain this issue in their discussion section. If the recording time is very short, it is impossible to obtain 70 trillion frames in one second in the real world.

[Response]: We agree with the reviewer that the recording time is an important parameter in ultrafast imaging. We also would like to point out that the maximum recording times for CUSP have been provided in the manuscript. In active mode with 70 Tfps, the maximum recording time is *14 ps* with a total of 980 frames. Please see Figs. 3b and 3c. In passive mode with 0.5 Tfps, the maximum recording time is *800 ps* with a total of 400 frames. Please see Fig. 4e. Note that since CUSP is a single-shot imaging technique, the recording time equals the actual duration that the observed ultrafast event occurs.

We also agree with the reviewer that there exists a trade-off between the time resolution and the recording time. However, we would like to point out that this trade-off is *universal*, which applies to all single-shot trillion-fps imaging modalities (e.g. references 12-19 in Main Text). This trade-off is inevitable since all these techniques rely on burst imaging. A general rule of thumb is that *the recording time is longer with a slower imaging speed*. As a result, it becomes

inconvenient and somewhat unfair to compare different techniques due to this inherent dependency between the recording time and the imaging speed. *Therefore, we had to introduce a parameter termed sequence depth, which is independent of the imaging speed and is defined as the total number of frames in one acquisition* (see the second paragraph in Main Text). This sequence depth parameter can provide a fairer quantitative comparison between different single-shot femtosecond imaging modalities. None of the previously constructed systems are able to offer sequence depth >350 frames and most of them are unfortunately limited to less than 10 frames. *As a new record, CUSP experimentally reached a sequence depth of 980 frames* (see Fig. 3), in addition to its record in imaging speed.

At the end of Methods in the revised manuscript, we added a new section talking about the upper limit on sequence depth. In brief, for a system with fixed hardware parameters, the sequence depth is determined by the desired field of view (FOV). *The larger the FOV, the shallower the sequence depth.* In Methods, we also wrote the equations to quantify the maximum sequence depth. We also added a new paragraph (the third paragraph) in the “Discussion” section in the revised manuscript.

Comment 1.3. In the discussion, it is better to add few sentences about the limitations of the technology. This current technology is probably only limited to relatively stationary samples (compared to the trillion frames per second speed). If your sample is moving (or shape changing) at the speed of your frame rate, is it possible to capture the above mentioned changes? Although the majority of samples do not move that fast (they are stationary sample in this time scale).

[Response]: We agree with the reviewer that it is necessary to talk about limitations of CUSP to provide a full picture of our technology since every technology has its limitations. Therefore, we added three new paragraphs in the “Discussion” section in the revised manuscript, where we discuss several technical aspects that we can potentially improve or overcome. *We combined comments from both Reviewer 1 and Reviewer 2 in these three paragraphs, including discussions in sequence depth (recording time), FOV, sparsity, spatial resolution, etc.*

However, we respectfully disagree with the reviewer that our technology is limited to relatively stationary samples. In fact, in our second set of experiments (Fig. 3), we observed ultrashort light pulse propagation inside a nonlinear optical medium. The light pulse is not

stationary but travels at the speed of light. The experiments on SR-FLIM (Fig. 4) indeed used stationary samples, meaning they did not move in space during acquisition, as the reviewer has noticed. However, the reason for using stationary samples is that for applications of SR-FLIM (also FLIM), *there is hardly any sample that moves or changes its shape at our imaging speed (0.5 Tfps)*, as the reviewer already mentioned. We would like to argue that *it is actually possible for CUSP to capture these changes, assuming that we could get such a sample which moves this fast*. Since CUSP and coded-aperture imaging share the same spirit of compressed sensing, we would like to resort to this paper for support, though their speed is much lower: *T-H. Tsai, P. Llull, X. Yuan, L. Carin, and D. J. Brady, "Spectral-temporal compressive imaging," Opt. Lett. 40, 4054-4057 (2015)*. In this work, the authors captured the movement of some colorful macroscopic objects in single-shot and resolved their spectra.

Comment 1.4. The authors should have used solvent effect to measure the lifetime testing rather than the concentration variation in Rh6G. The concentration used was very high so low lifetime values were recorded. Any new development should be compared with the existing literature value <https://pubs.acs.org/doi/pdf/10.1021/j100535a016>

In addition, they have not compared the measured lifetime with the TCSPC or any other existing methods to prove that their measurement is reproducible with the literature value.

[Response]: We thank the reviewer for insightful suggestions. The reason why we used concentration instead of solvent effect in lifetime testing is that we can *study the trend of lifetime against concentration* (Supplementary Fig. S13c). Usually fluorescence lifetime is independent of concentration. However, this common observation only holds at low concentrations when non-radiative decays are negligible. *At high concentrations, non-radiative decays start to rise and promote the fluorophores to decay faster*. We studied this counter-intuitive phenomenon by measuring lifetimes at various fluorophore concentrations.

The fluorophore concentration in our study fall out of the range that was measured in the paper pointed out by the reviewer (added as reference 38 in Main Text). However, *our lifetime versus concentration plot* (Supplementary Fig. S14c) *follows the same trend as that demonstrated in that previous work*. In addition, the temporal resolution of the measurement system in reference 38 is only 1 ns, which is on par with or even longer than the measured

lifetimes, therefore their data is unable to contribute to an accurate quantitative comparison. Therefore, we would like to claim that passive CUSP with a temporal resolution of 20 ps can actually quantify lifetimes more accurately.

In order to compare our single-shot SR-FLIM results with *an existing standard method*, we directly measured Rh6G solution lifetime using only the streak camera in conventional mode. This method has been ubiquitously employed in the literature (see the newly added references 40-42 in Main Text). In this reference experiment, we used uniform samples and imaged them using the streak camera without going through the CUSP system. A narrow slit (40 μm wide) was applied at the input of the streak camera. Eventually, we can obtain the lifetime values via exponential fit on the temporal profile of fluorescence emission intensity. The results are summarized in Supplementary Figs. S14d and S14e. The reconstructed lifetimes from passive CUSP match well with the reference data from the direct streak camera measurements. *The average error is only 10 ps.*

To accommodate these contents, we added a new paragraph in the “SR-FLIM” section in the revised manuscript and also added a new paragraph in Supplementary Note 9.

Reviewer 2

Comment 2.1. The manuscript by Wang et al reports on a photography technique capable of capturing a large number of frames at a high frame rate of a single event in the femtosecond regime. The authors break records in number of frame rate and number of frames. There is no doubt that the experiment is nice and well done.

[Response]: We thank the reviewer for acknowledging that our work *breaks records in both frame rate and number of frames* and is *well-done*.

Comment 2.2. The manuscript is in general extremely emphatic (which is quite annoying) without explaining the science behind. As an example: the abstract even doesn't mention the key concept which enabled the record. Other example: "One promising approach is CUP which creatively combines streak camera and compressed sensing" (with 2 references to the authors own work). Identically, the conclusion extends the possibilities of this techniques to fields that are quite exaggerated: even I must admit that it might be possible one day, the path is quite long to detect gravitational waves from colliding black holes using CUSP...

[Response]: Although we respectfully disagree on this comment, we'll do our best to mitigate the concern.

- 1) In our abstract, the key concepts enabling the new records are in fact all listed in this sentence: "*In active mode, CUSP achieves both 7×10^{13} fps and 10^3 frames simultaneously by synergizing spectral encoding, pulse splitting, temporal shearing, and compressed sensing—enabling...*" This is also affirmed by the comments from Reviewer 3.
- 2) In addition, following that sentence that introduces CUP, we do have two sentences that describe the working principles of CUP. In the revised manuscript, we have added some more information.
- 3) We understand the concern of the reviewer that there is a long way to go before CUSP can be deployed in these exciting fields. However, we list these potential applications in the "Discussion" section to showcase *the benefit of single-shot ultrafast imaging compared to the traditional pump-probe-based methods*. In order to avoid confusions, we added a sentence in the "Discussion" section in Main Text: "*Large amounts of theoretical*

and technical efforts are required before CUSP can be widely adapted for these applications.” In fact, we have been in direct contact with experienced astronomers and astrophysicists who are interested in collaboration to incorporate our technology to study astronomical events.

Comment 2.3. As a consequence of this lack of simple scientific explanations, it is to me hard to find where is the actual new concept in comparison with reference 14 (Lu et al, Phys. Rev Lett 2019 Figure 1c.)

[Response]: We thank the reviewer for directing us to this work in reference 14. We can now explain how CUSP is substantially different and advantageous.

- 1) Reference 14 only uses the spectrum of one stretched pulse to stamp time. It is conceptually a “compressed” version of STAMP (see reference 12). Therefore, their sequence depth (defined as the number of captured frames in each acquisition) is only 60 frames. *CUSP leverages two more concepts: pulse splitting and temporal shearing, in addition to the concepts of spectral encoding and compressed sensing.* Only the last two were used in reference 14. Therefore, *this innovation in concept synergy elevates our sequence depth to 980 frames*, which is more than an order of magnitude beyond that in reference 14. As mentioned in our response to Comment 2.2, this synergy of four concepts can be found in Abstract. Please refer to the “Principles” section, Fig. 1, Methods, and Supplementary Note 3 for details in the implementations of pulse splitting and temporal shearing.
- 2) Spectral bandwidth of an ultrashort pulse and its temporal duration is related via Fourier transformation. In addition, temporal chirp is required for time stamping (see reference 12). Therefore, there is always an optimal operation point that offers the best temporal resolution, which was analyzed in detail in reference 13. *However, reference 14 failed to operate at this optimal condition, so their temporal resolution (a few ps) is much worse than that of active CUSP (240 fs, see Fig. 2c).* Please see our response to Comment 2.10 for more explanations on this optimal temporal resolution.
- 3) Without the concept of temporal shearing, reference 14 unfortunately is unable to realize single-shot ultrafast spectral imaging (x, y, t, λ) . Their temporal sequence of spectral

images (Fig. 3 in reference 14) were still acquired via the traditional “pump-probe” scheme, which requires multiple shots while varying the time delay between the pump and probe beam. On the contrary, our CUSP in passive mode is capable of achieving truly single-shot ultrafast spectral imaging – *we can observe spatio-spectrally resolved fluorescence decays in real-time at 0.5 Tfps*. This advantage gives us significant edge in both measurement throughput and accuracy.

Comment 2.4. The authors always avoid quantitative comparisons. As a scientist, I know that nothing is magic and that incredible numbers here are always linked to some losses there (in resolution, information, ...etc). It might be acceptable, but the manuscript doesn't describe them at all. Apart from the records numbers, it is in general quite difficult to get factual numbers or to see from where they are coming from. Examples: what are the temporal and spatial resolutions? The authors seem to always avoid this point:

i /it is never provided in the main text

ii/ in Supplementary material page 3 “with a moderate SNR, the temporal resolution is typically larger than 400 fs [2]. Fortunately, the temporal resolution of active CUSP is no longer bounded by this limit.” : so... what is it?

iii/ In Suppl. Mat page 16 “After CUSP reconstruction, we spatially integrated the reconstructed intensity and plotted it in the t - λ space (Supplementary Fig. S13a). It is the impulse response of the system, or in other words, the 2D PSF. The 1D PSFs in the spectral and time domains give FWHMs of 13 nm and 20 ps, respectively”. So that the reader has the 1D PSF in the case of passive CUSP. It is not mentioned for the interesting (central in the manuscript) case of the active one. (Besides, I hope that it is much shorter than 20 ps....)

[Response]: We agree with the reviewer that it is necessary to provide quantitative numbers when evaluating the performances of a new imaging system.

- 1) However, we respectfully disagree since *the temporal resolution of active CUSP (240 fs) was explicitly given in Line 120 and Fig. 2c in the original manuscript*. It is defined (and also was measured) as FWHM of the intensity profile at a spatial location when the spatio-temporally chirped pulse train sweeps across a planar sample. This method of temporal resolution characterization was also used in previous publications such as

reference 18. To make it clear, we added one sentence in this paragraph in the revised manuscript. We also added this resolution number in the last paragraph of Supplementary Note 1 in the revised manuscript. As for passive CUP, the reviewer is right that the demonstrated temporal resolution is ~ 20 ps, as explicitly studied in Supplementary Note 9 and Supplementary Fig. S14a.

- 2) As for spatial resolutions, we thank the reviewer for this insightful suggestion. In the revised manuscript, we characterized the spatial resolutions of both active and passive CUSP by imaging a standard pattern and then Fourier-transforming the spectrotemporally integrated CUSP data (see the newly added Supplementary Note 10 and Supplementary Fig. S15). *It shows that spatial resolutions are isotropically decreased by $2.8\times$ and $2.1\times$ in active CUSP and passive CUSP, respectively. Degradation in spatial resolution is one of the most prevalent and pressing issues in all compressed-sensing-based imaging modalities due to multiplexing of information from other dimensions into the spatial domain (see references 7, 8, 17-22 in Supplementary Information).* Lossless encoding and multi-view projection may be promising candidates to improve spatial resolutions. We commented on spatial resolutions in the “Discussion” section in the revised manuscript.

Comment 2.5. What is the field of view? How is it limited?

[Response]: We thank the reviewer for this useful suggestion. The field of view (FOV) of all three sets of experiments are: $12.13\text{ mm} \times 9.03\text{ mm}$ (Fig. 2), $2.48\text{ mm} \times 0.76\text{ mm}$ (Fig. 3), and $180\text{ }\mu\text{m} \times 180\text{ }\mu\text{m}$ (Fig. 4). We explicitly listed these numbers in the revised manuscript.

Factors that dictate the maximum FOVs include the mode of operation, the streak camera’s pixel counts, the streak camera’s pixel size, the number of frames, the number of wavelength samples, and magnification factor of the imaging optics. We developed a full set of equations to calculate the maximum FOVs at the end of Methods in the revised manuscript.

Comment 2.6. Last but not least example is to obtain 980 frames with only 7 pulses. This number of 980 would largely deserve some lines of comments to explain how it has been obtained and if it actually means something physically. (pixelation vs resolution)

[Response]: We thank the reviewer for pointing this out. Basically, this 980 is calculated by this equation $N_{\text{ta}} = PB_i|\mu|/d$, given in the third paragraph in the “Principles” section. Here, B_i (the bandwidth of the illumination light pulse) multiplied by $|\mu|$ (the spectral dispersion parameter of the system) represents the spatial extension in the x_s direction due to the diffraction grating. This number divided by d (the streak camera’s pixel size) gives the number of pixels the spectral dispersion extends, which equals the number of reconstructed frames from a single sub-pulse (i.e. 140 frames here). Multiplying this with P (the number of used sub-pulses, $P = 7$ here) results in the total number of frames. This can be understood by referring to the top right inset of Fig. 1. An alternative way of calculating this number is by *multiplying the total observation window of 14 ps by the frame rate of 70 Tfps*. For clarity, we added a brief explanation in the second paragraph in the “Imaging an ultrafast nonlinear optical phenomenon” section in the revised manuscript.

Comment 2.7. In average, the main text is not understandable by itself without reading the Supplementary material. As an example, the actual role of the DMD is not described in the main text and barely mentioned in the suppl. mat.

[Response]: Actually, the role of DMD was described in the first paragraph of the “Principles” section by this sentence: *“In the other path, the image is encoded by a digital micromirror device (DMD), displaying a static pseudo-random binary pattern, ...”* Basically, DMD was used to spatially encode the imaged dynamic scene. The encoding pattern is a computer generated pseudo-random binary pattern, like a QR code. Spatial encoding by either a DMD or photomask is a standard method in compressed-sensing-based imaging, such as reference 14 and coded aperture imaging: *T-H. Tsai, P. Llull, X. Yuan, L. Carin, and D. J. Brady, “Spectral-temporal compressive imaging,” Opt. Lett. 40, 4054-4057 (2015)*. Additionally, we used a whole paragraph (the third paragraph) in the “Equipment” section in Methods to explain the working principle of DMD spatial encoding. To provide more information, we added a sentence in the first paragraph in the “Principles” section in the revised manuscript.

Comment 2.8. Line 67: it is hard to see what are these passive/active modes before understanding the initial concept.

[Response]: We thank the reviewer for careful reading. To make it clear, we enriched the contents in the first paragraph in the “Principles” section in the revised manuscript.

Comment 2.9. Line 103: I do not see how the frame rate R_p is calculated since each 2D frame is not with a single pixel.

[Response]: We thank the reviewer for this comment. The calculation of frame rate R_p is based on the working principle of the streak camera. Inside the streak camera (see Methods and Supplementary Fig. S1), the accelerated photoelectrons are deflected by a rapid ramping voltage. The sweeping speed is denoted as v in our manuscript. As a result, the streak camera converts the temporal evolution of a dynamic scene into spatial shearing of the scene in the vertical direction (see the top right inset of Fig. 1 and Supplementary Fig. S1). In other words, *each frame is shifted in the streak camera by one pixel in the vertical direction (y_s). Therefore, the temporal spacing between two adjacent frames is d/v , indicating a frame rate of $R_p = v/d$.* To justify this calculation, we added a sentence in the third paragraph of the “Principles” section in the revised manuscript.

Comment 2.10. Lines 119-122 are not understandable without reading ref 13

[Response]: We apologize for the lack of information for the reviewer to grasp our contents. We added several sentences at the end of this paragraph to make it more understandable. Essentially, *both time-bandwidth limit and temporal chirp play roles in determining the optimal operation point.* In the high-spectral-resolution regime, time-bandwidth limit is the major contributor, so that higher spectral resolution leads to poorer temporal resolution. In the low-spectral-resolution regime, the effect of temporal chirp starts to become dominant, so that lower spectral resolution also leads to lower temporal resolution. *Thus, there is usually a sweet spot for spectral resolution that offers the best temporal resolution, for a given system with fixed hardware parameters.*

Comment 2.11. How is implemented T-CUP in the experiments? (removal of the grating?)

[Response]: We thank the reviewer for this good suggestion. To explicate how T-CUP was implemented in our system, we added Supplementary Note 7 and Supplementary Fig. S10 in the revised manuscript. We also added a sentence in the second paragraph in the “Imaging an ultrafast linear optical phenomenon” section in the revised manuscript.

The reviewer is correct that we *removed the grating* in front of the streak camera to implement T-CUP. However, that is not enough. Due to the diffraction angle by the grating, the original beam-steering mirror after the grating is no longer at the right position and the right angle to direct the light to the streak camera for T-CUP imaging. One simple solution is to place the grating and the mirror for CUSP on a *kinematic magnetic mount*, and place another mirror on another magnetic mount for T-CUP. Both magnetic mounts are coupled with a common magnetic base fixed on the optical table. In addition, we also had to adjust the position of the streak camera in the direction of light propagation by a few millimeters. All the other optical components remain unchanged. *This system design allows minimum changes in the experimental setup and easy transition between CUSP and T-CUP.*

Comment 2.12. Line 129: I do not see how the 7x increase comes from the figure.

[Response]: We thank the reviewer for this comment. In the revised manuscript, we added a new label in Fig. 3c, so that this 7× increase is visually more obvious.

Comment 2.13. Fig3a: why is P1 at 45° from the polarization axis of the Kerr gate? (it should be in the same direction?)

[Response]: We respectfully disagree since polarizer P1 should be at 45° in respect to the polarization axis of the Kerr gate. This is determined by the working principle of the Kerr gate. The gate pulse modulates the refractive index of the Kerr medium along its polarization direction, inducing instantaneous birefringence. After going through the first polarizer P1 (45°), the detection light is converted into elliptically polarized light. After passing through the second polarizer P2 (-45°), we can get a gate transmission, which is a function of gate pulse intensity. Please refer to Supplementary Note 8 for mathematical description of this process.

This configuration of a Kerr gate has been extensively exploited in ultrafast imaging. Please see reference 32 and this early work from Bell Lab: *M. A. Duguay and A. T. Mattick, "Ultra-high Speed Photography of Picosecond Light Pulses and Echoes," Appl. Opt. 10, 2162-2170 (1971)*. We added this paper as reference 31 in the revised manuscript.

If P1 was aligned in the same direction with the polarization axis of the Kerr gate, the linearly polarized light after P1 could not be converted to elliptical polarization so that the induced birefringence would be useless.

Comment 2.14. Line 157: how are obtained the 980 frames? I do not see how this number can be obtained with only 7 laser pulses. Are they “physically” independent? i.e. separated by the temporal PSF?

[Response]: Please see our response to Comment 2.6 about how this was calculated. These 7 sub-pulse can be considered as “physically independent”, separated by t_{sp} (please see Fig. 1, and our response to Comment 2.26).

Comment 2.15. Line 158: the long paragraph is nice, but the only number which would interesting (the relaxation time that could be compared with the experimental data) is not provided.

[Response]: We thank the reviewer for this advice. We estimated the relaxation time of BGO in the revised manuscript. Since Fig. 3f is a convolution of the Kerr effect relaxation and the CUSP’s temporal PSF (Fig. 2c), we can derive *a relaxation time of 380 fs by deconvolution*. We added this in the third paragraph in the “Imaging an ultrafast nonlinear optical phenomenon” section.

Comment 2.16. Line 172: why an amplification of the noise by 11 times for a Kerr process? Comment is needed.

[Response]: The analysis of noise amplification was detailed in Supplementary Note 8 in the revised version. Basically, *the transmittance of a Kerr gate is a nonlinear function of the induced*

phase retardation and thus a nonlinear function of the gate pulse intensity [see Equation (S35)]. As a result, we can derive a nonlinear function describing how fractional change of the Kerr gate transmittance is related to the fractional change in the gate pulse intensity [see Equation (S37) and Supplementary Fig. S12a]. In our experiment, *we designed a highly sensitive Kerr gate system that operates at 0 to $\pi/9$ phase retardation, where noise amplification is large.* Employing such an unstable “chaotic” system, we demonstrated the necessity and advantage of single-shot ultrafast imaging compared to the traditional pump-probe technique that requires event repetition (Supplementary Movie 3). We added an explanation in the last paragraph in the “Imaging an ultrafast nonlinear optical phenomenon” section in the revised manuscript.

Comment 2.17. Line 191: the lifetime should be provided quantitatively and compared to literature on a quantitative basis.

[Response]: We thank the reviewer for this helpful advice. Please see our response to Reviewer 1’s Comment 1.4.

Comment 2.18. *Can the sparsity be justified?*

[Response]: We thank the reviewer for this insightful comment. In the revised manuscript, we calculated the sparsity of the transient scenes imaged by CUSP as summarized in Supplementary Table 1. Here we define sparsity as the number of voxels with intensity below the standard deviation of the background noise divided by the total number of voxels in the reconstructed dataset. *All our experiments have data sparsity greater than 90 %.* Based on the analysis presented in reference 52 in Main Text, 90% sparsity (or equivalently 10% data density) can lead to reasonably good resolutions in compressed-sensing-based single-shot imaging. The definition of sparsity can be found here: https://en.wikipedia.org/wiki/Sparse_matrix. We also added this information in the “Discussion” section in the revised manuscript.

Comment 2.19. What is the exact mathematical mean of the argmin function and the kind of norm 2? (these should be specified clearly)

[Response]: In Equation (2), “argmin” means *arguments of the minima*. It is defined as the point in the multi-dimensional variable space where the function in the following bracket $\{\}$ has minimal values. This notation is extensively used in mathematics. Please see this reference: https://en.wikipedia.org/wiki/Arg_max.

Here, we used L^2 norm (or Euclidean norm), defined as $\|x\|_2 = \sqrt{\sum_{k=1}^n |x_k|^2}$, in which x is an n -element vector $x = [x_1, \dots, x_n]$. L^2 norm is extensively used in mathematics and engineering. Please see these references: [https://en.wikipedia.org/wiki/Norm_\(mathematics\)](https://en.wikipedia.org/wiki/Norm_(mathematics)) and <http://mathworld.wolfram.com/L2-Norm.html>. We added some explanations in Methods in the revised manuscript.

Comment 2.20. In general, the supplementary document strongly lacks of referencing. Yet it is detailed, it would be good to know how the formulas have been established (example: equation S2 for the transformation D).

[Response]: We thank the reviewer for careful reading. D describes the *transformation from the image in u -View to that in s -View*. It is essentially a 3×3 matrix and we denote it as Equation (S2) in order to include all geometric factors that contribute to this transformation. We derived Equation (S2) from the standard camera calibration matrix in photography. Please see the following references: Z. Zhang, “Flexible camera calibration by viewing a plane from unknown orientations,” *Proceedings of the Seventh IEEE International Conference on Computer Vision (Kerkyra, Greece), 1*, 666-673 (1999), and <https://www.mathworks.com/help/vision/ug/camera-calibration.html>. We added a sentence and these references after Equation (S2) in Supplementary Note 2 in the revised manuscript.

In addition, we also added more references in Supplementary Information in response to the reviewer’s other comments.

Comment 2.21. It would be help to tell the reader where the characterizations have been performed without the DMD and retrieval algorithm (if so).

[Response]: We thank the reviewer for this helpful suggestion. The quantum efficiency data (Supplementary Fig. S2a) and the temporal resolutions of the streak camera at different light

levels were provided by the streak camera's manufacturer – Hamamatsu Corp. When characterizing the space-charge effect and response linearity, we illuminated the DMD with a uniform beam from the femtosecond laser. The DMD displayed a static pattern (see the insets of Supplementary Fig. S2b) and the reflected light from the DMD is relayed to the streak camera. No grating was present before the streak camera. Then we acquired data while varying the intensity of the illumination beam. We added a paragraph explaining these characterization experiments in Supplementary Note 1 in the revised manuscript.

Comment 2.22. Spectral sensitivity: the authors assume uniform transmission of the optics etc, but in fact, this should be already taken into account in the quantum efficiency of the apparatus.

[Response]: We respectfully disagree. The photocathode's quantum efficiency plot in Supplementary Fig. S2a was measured without the input optics of the streak camera. We specified this information in the revised manuscript.

Comment 2.23. What is the NA of the optics? What is the resolution necessary to image the DMD?

[Response]: The NA of the relay optics that images the DMD to the streak camera is ~ 0.08 . Using a central wavelength of 800 nm, it gives a spatial resolution of 5 μm . The lateral dimension of one DMD code in the pseudo-random pattern is 56.7 μm and the magnification from the DMD to the streak camera is $1\times$. *Therefore, the relay optics has enough spatial resolution to image the DMD.* We added a sentence in the third paragraph of the “Equipment” section in the revised manuscript.

Comment 2.24. What is “rect” function precisely standing for?

[Response]: In the science and engineering communities, “rect” typically represents a rectangular function. Its definition in 1D is here: https://en.wikipedia.org/wiki/Rectangular_function. In our work, we used its 2D version, defined as:

$$\text{rect}(x, y) = \begin{cases} 1, & \text{if } |x| \leq \frac{1}{2} \text{ and } |y| \leq \frac{1}{2} \\ 0, & \text{if } |x| > \frac{1}{2} \text{ or } |y| > \frac{1}{2} \end{cases}$$

We used this 2D “rect” function to represent one camera’s physical pixel. We added one sentence and a reference right after Equation (S7) in the revised manuscript.

Comment 2.25. Page 9: it would be good to make the link with the operator O used in the main text

[Response]: We added one sentence after Equation (S20) to make a connection between the operator O and the equations in Supplementary Note 4.

Comment 2.26. What is precisely t_{sp} ?

[Response]: Here, $t_{\text{sp}} = 2$ ps represents the temporal separation between sub-pulses, as defined in the second paragraph of the “Principles” section and in Fig. 1. The setup that generates these sub-pulses is detailed in Supplementary Note 3 and Supplementary Fig. S6.

Comment 2.27. Page 11: how the regularization parameters were chosen? Impact on results?

[Response]: We thank the reviewer for careful reading. *The regularization parameter balances the two terms in Equation (2) and promotes convergence to the best solution for an inverse problem.* With too small a parameter, the result is overwhelmed by noise; while with too large a parameter, the result becomes over-smoothed. Therefore, a right regularization parameter has to be chosen. In our implementation, *we opted to use trial-and-error to select the regularization parameter.* We inspect the reconstruction results while varying the parameter value. We added one sentence in Supplementary Note 5 in the revised manuscript.

Comment 2.28. Page 17: order of magnitude for the decay? (so that one can compare to the results)

[Response]: Lifetimes of commonly used fluorophores typically range from hundreds of picoseconds to tens of nanoseconds. We added one sentence and a reference in the Supplementary Note 9 in the revised manuscript.

Reviewer 3

Comment 3.1. This paper proposes a new ultrafast imaging technique that significantly extends the previous state of the art in several directions. The proposed method leverages several techniques (namely compressive sensing, spectral encoding, pulse splitting, and temporal shearing) to achieve impressive in ultrafast imaging. Moreover, the proposed approach is flexible in that it allows a tradeoff between speed and spectral resolution, making it also an ultrafast multispectral imaging technique. There are many applications for such fast imaging capabilities, thus these advances are not only technically, but also scientifically very relevant. As far as I am aware, this is a qualitative jump in the field of ultrafast imaging, thus it is an important contribution to the toolbox of science.

The paper is well written, the technique clearly motivated and explained, thus I believe that this paper is worthy of publication in its current form.

[Response]: We thank the reviewer for supporting our work and acknowledging that our work is *a qualitative jump in the field of ultrafast imaging* and is *an important contribution to the toolbox of science*. We have further improved our manuscript by careful revision based on the comments from the other reviewers.

REVIEWERS' COMMENTS:

Reviewer #1 (Remarks to the Author):

I am satisfied with the answers and the appropriate correction in the manuscript. It will be an useful article for the readers of this journal. I recommend publication of this interesting work.

Reviewer #2 (Remarks to the Author):

The authors have correctly amended the manuscript. They have answered relatively precisely to all questions raised by the referees.

In my opinion, because a journal is intended for wide audience, the new paragraph in the discussion still lacks of clarification about orders of magnitude.(but at least, this time, they have been provided here and there in the manuscript).

2 minor comments:

- t_{sp} is not defined on line 93 (it is defined only from line 117)

-it is somehow surprising that quantitative values for field of view are given only for SR-FLIM. Is the value close for the active case ? If not, it would be interesting to know why.

Response to Reviewers

We thank the reviewers for their insightful comments, which have helped us improve the quality of our manuscript. Below, we provide point-by-point responses to all the comments.

Reviewer 2

Comment 2.1. The authors have correctly amended the manuscript. They have answered relatively precisely to all questions raised by the referees.

[Response]: We thank the reviewer for acknowledging that we have correctly amended the manuscript.

Comment 2.2. In my opinion, because a journal is intended for wide audience, the new paragraph in the discussion still lacks of clarification about orders of magnitude. (but at least, this time, they have been provided here and there in the manuscript).

[Response]: We respectfully disagree with the reviewer. Clear quantifications of our CUSP system are given in the discussion paragraphs, including sequence depth, spatial resolution, and sparsity. Other quantifications, such as temporal resolutions and FOVs, are presented in other paragraphs in the manuscript.

Comment 2.3. - t_{sp} is not defined on line 93 (it is defined only from line 117)

[Response]: We respectfully disagree with the reviewer. Before line 93, t_{sp} is already defined in lines 89-90: "...a pulse train with neighboring sub-pulses separated by time t_{sp} , ...". Here, the exact value of t_{sp} in our experiment is not given since the purpose of this paragraph is to describe the working principle of our system and define its parameters. In line 117, the exact value of t_{sp} is given when we show our experiments and results in detail.

Comment 2.4. -it is somehow surprising that quantitative values for field of view are given only for SR-FLIM. Is the value close for the active case? If not, it would be interesting to know why.

[Response]: We respectfully disagree with the reviewer. The field of views (FOVs) for all three experiments were provided in our manuscript. The FOV of the first experiment “imaging an ultrafast linear optical phenomenon” was given in line 119: “Exemplary frames from CUSP reconstruction with a field of view (FOV) of 12.13 mm×9.03 mm are summarized in Fig. 2b.” The FOV of the second experiment “imaging an ultrafast nonlinear optical phenomenon” was given in line 163: “..., the gate focus was outside and inside the FOV (2.48 mm×0.76 mm in size), respectively.” The FOV of the third experiment “SR-FLIM” was given in line 207: “The final data has an FOV of 180 μm×180 μm, ...”

In addition, we developed a full set of equations to calculate the maximum FOVs at the end of Methods. Factors that dictate the maximum FOVs include the mode of operation, the streak camera’s pixel counts, the streak camera’s pixel size, the number of frames, the number of wavelength samples, and magnification factor of the imaging optics.